# Assessing afebrile malaria and bed-net use in a high-burden region of India: Findings from multiple rounds of mass screening

**Samir Garg**[1]*, **Vishnu Gupta**[1], **Kavita Patel**[1], **Mukesh Dewangan**[1], **Prabodh Nanda**[1], **Ryavanki Sridhar**[2], **Gajendra Singh**[2]

**1** Division of Health Systems, State Health Resource Centre, Raipur, Chhattisgarh, India, **2** Health Section, UNICEF, Chhattisgarh Field Office, Raipur, India

* koriya@gmail.com

**Data Availability Statement:** All relevant data are within the paper and its Supporting information files.

## Abstract

A key obstacle in the fight against malaria is afebrile malaria. It remains undiagnosed and, therefore, is invisible to the health system. Apart from being a serious illness, it contributes to increased transmission. Existing studies in India have not adequately reported afebrile malaria and its determinants, including the use of long-lasting insecticide-treated nets (LLINs). This study used six waves of mass screening, which were conducted by the state government in the high-malaria-burden region of Chhattisgarh, a state in India, in 2020, 2021, and 2022. Each round of data collection included more than 15000 individuals. Descriptive statistics were used to analyse key indicators of malaria prevalence and LLIN use. Multivariate analyses were performed to identify the determinants of afebrile malaria and LLIN use. Malaria prevalence in the afebrile population varied from 0.6% to 1.4% across the different waves of mass screening. In comparison, malaria positivity among febrile individuals was greater than 33% in each wave. Afebrile malaria contributed to 19.6% to 47.2% of the overall malaria burden in the region. Indigenous communities (scheduled tribes) were more susceptible to malaria, including afebrile malaria. Individuals using LLINs were less likely to be affected by afebrile malaria. Overall, 77% of the individuals used LLINs in early monsoon season, and in winter the rate was lower at 55%. LLIN use was significantly associated with the number of LLINs the households received from the government. Although fever continues to be a primary symptom of malaria, afebrile malaria remains a significant contributor to the malaria burden in the region. The free distribution of LLINs should be expanded to include high-burden populations. Global policies must include strategies for surveillance and control of afebrile malaria in high-burden areas.

## Introduction

The elimination of malaria is a global priority under the Sustainable Development Goals [1]. The Global Technical Strategy for Malaria Elimination 2016–2030 lays out specific goals, milestones, and targets for eliminating malaria by 2030 [2]. In 2019, the World Health Organization (WHO) launched the high burden to high impact (HBHI) strategy as a country-driven

**Funding:** UNICEF Field Office, Chhattisgarh, India. The funders had no role in study design and analysis. The funders were involved in the decision to publish and in preparation of the manuscript.

**Competing interests:** The authors have declared that no competing interests exist.

response to have a rapid and sustainable impact on malaria. The HBHI strategy aims to use accessible and affordable frontline services to accelerate progress in countries with the highest disease burden [3]. Malaria accounted for an estimated 241 million infections and 627000 deaths globally in 2020 [1]. Although the prevalence of malaria has declined, many parts of Africa and Southeast Asia continue to experience a high malaria burden. In 2020, India accounted for 83% of the malaria cases in Southeast Asia [1]. Additionally, India is one of 11 countries where the HBHI approach has been advocated by the WHO. Some remote and forested region states in India, such as Odisha, Chhattisgarh, and Jharkhand account for a major share of the malaria burden [4, 5].

The Global Technical Strategy for Malaria Elimination 2016–2030 has identified asymptomatic malaria as a major challenge because, such infections are invisible to the health system and tend to remain undiagnosed [2]. People with asymptomatic infections are those who have parasites for a long time but do not have any clinical symptoms of the disease and have not recently been treated with antimalarial medications [6]. Asymptomatic infections contribute significantly to the malaria transmission cycle. Asymptomatic malaria infections are also associated with recurrent episodes of symptomatic parasitaemia, chronic anaemia, maternal and neonatal mortality, coinfection with invasive bacterial diseases, and cognitive impairment [7].

There is a significant presence of asymptomatic malaria in malaria-affected regions worldwide. A study in Ethiopia in 2018 reported the prevalence of malaria among asymptomatic individuals as 4.8% using rapid diagnostic tests (RDT) and 4.2% using microscopy [8]. A study using RDT in the Central African Republic in 2021 revealed malaria positivity in 51.2% of children and 12.2% of asymptomatic adults [9]. A study using polymerase chain reaction (PCR) tests in south-eastern Bangladesh discovered that 77% of malaria cases between 2007 and 2008 were asymptomatic infections [10]. A large-scale microscopy-based study in Cambodia reported that 67%–90% of the malaria patients did not have fever [11].

In India, a 2011 study using microscopy in the state of Maharashtra found that 4.28% of asymptomatic individuals tested positive for malaria [12]. A 2012 study from West Bengal, using a combination of PCR, microscopy, and RDT, found 8.4% of the asymptomatic population infected with plasmodium falciparum [13]. A 2014 study using microscopy in central India revealed that the prevalence of malaria among asymptomatic individuals was approximately 20% [14]. A 2017 study in Odisha found that 57% of the plasmodium falciparum cases detected by PCR were asymptomatic [15]. The aforementioned studies on asymptomatic and afebrile malaria in India had small sample sizes. The studies were not representative of any particular region, as they were limited to a few villages or a single district. Since 2017, not many studies have been conducted on asymptomatic or afebrile malaria cases in India. A recent study of asymptomatic malaria in Chhattisgarh focused exclusively on pregnant women [16].

None of the existing studies in India has examined the determinants of asymptomatic malaria, including the use of mosquito bed nets or long-lasting insecticide-treated nets (LLINs). It is globally acknowledged that LLINs offer essential protection against mosquitoes and significantly reduce malaria-related morbidity and mortality, particularly in endemic areas [17–19]. Results from a meta-analysis of African countries found that 72.1% of households and 52.5% of the individuals owned bed nets. The rate of bed-net use was 41.2% [20]. In India, a 2014 study on 15 villages in Odisha found that the rate of LLIN-use varied between 57.9% and 90.2% [21]. A 2015 study in a subdistrict of Chhattisgarh reported that 59.4% of individuals had used LLINs the previous night [22]. A study in Meghalaya in 2019–20 revealed that 62.2% to 100% individuals used bed nets [23]. However, none of the above studies examined the factors associated with the use of LLINs in India.

Malaria is endemic to Chhattisgarh, India. Chhattisgarh is one of the four high-malaria burden states in India selected for the HBHI initiative [3]. The state has 44% of its area covered by forests [24]. The incidence of malaria in the state varied widely across districts. For example, in 2018, the annual parasite index (API) of Raipur district in central Chhattisgarh was 0.02 whereas it was 53.08 in Bijapur district in southern Chhattisgarh. The state has five geographical divisions, and the Bastar division accounted for the bulk of the malaria burden in 2018 [25, 26].

In 2019–20, the Chhattisgarh government decided to adopt a strategy of Mass Screening and Testing (MSaT) to combat malaria in Bastar. This strategy involved testing the entire population in the high-burden region using RDTs and treating all malaria-positive individuals. RDTs are lateral-flow immunochromatographic antigen detection tests. In 2016, the neighbouring state of Odisha implemented the MSaT approach in high-burden districts [27]. In the Bastar region of Chhattisgarh, six rounds of MSaT were conducted between January 2020 and June 2022 with the aim of covering a population of more than one million in each wave. These rounds of the MSaT offered the opportunity to conduct a study on asymptomatic malaria with a large sample size in the region. Given that fever is the most well-known symptom of malaria, this study focused on afebrile malaria, which is the predominant type of asymptomatic malaria. The following were the specific objectives of this study:

1. To examine the prevalence of afebrile malaria and its determinants

2. To determine the rate of LLIN use by individuals and its determinants

## Materials and methods

### Study setting

This study was conducted in the southern part of Chhattisgarh, a state in India. It is a contiguous area in the Bastar division, with an API above 20 in 2018. The geographical setting of the study is shown in Fig 1, and was created using an open-source desktop geographical information system software application called MAPWINDOW (version 5) [28]. This area shares borders with the states of Odisha, Andhra Pradesh, Telangana, and Maharashtra. It is one of the poorest regions in India [29]. The population of the area is approximately one million, and vulnerable indigenous communities, categorised as scheduled tribes, constitute a major share.

### Study design

This study analysed six repeated cross sections corresponding to the six waves of the MSaT implemented in 2020, 2021, and 2022. It was implemented mainly through community health workers (CHWs) known as 'mitanins', supervisors of CHWs, and government paramedical workers trained to use the RDT. The CHWs had around five years of experience in testing febrile individuals for malaria using bivalent RDTs and treating malaria cases at the community level [29]. The results of each individual's malaria tests during the MSaT were recorded by the aforementioned health workers in the format prescribed by the state government. The participant's temperatures were taken at the time of testing during the MSaT, and information on the presence or absence of fever was recorded using the prescribed format.

The RDTs employed were approved by the WHO for use under field conditions. They have a false-positive rate of 1.4% and 1.3% for Plasmodium falciparum (Pf) and Plasmodium vivax (Pv), respectively [30].

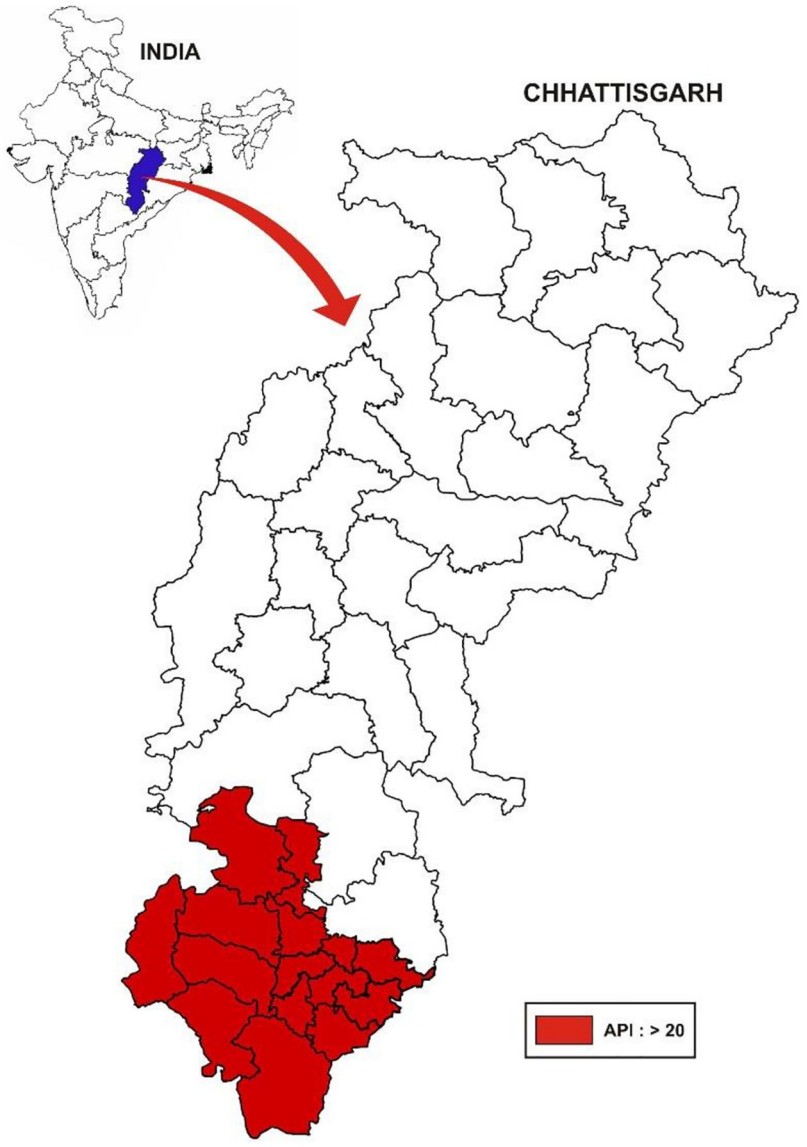

**Fig 1. The study area.**

Six rounds of MSaT were conducted in February 2020, July 2020, February 2021, July 2021, December 2021, and June 2022. The completion of each round of MSaT was followed immediately by a wave of household surveys completed within a month.

## Sampling

This study compared malaria prevalence between febrile and afebrile individuals. The minimum sample size required to yield an adequate number of febrile individuals was calculated. A minimum number of 385 febrile individuals were needed for 5% precision at 95% confidence level (CI), for which it was estimated that around 15,000 individuals would have to be covered. A large household survey was conducted, with each wave covering more than 15,000 individuals. The study area (Fig 1) consists of 15 administrative units known as blocks. Eight

habitations were selected through systematic random sampling of the list of habitations in each block. Thus, 120 habitations were selected. It was designed to cover all households in the selected habitations. All individuals in each household were covered.

### Data collection

A structured questionnaire was used to interview the individuals. It contained questions on people's socio-demographic characteristics, availability and use of LLIN, whether they were tested for malaria during the MSaT, whether they had fever on the day of testing, and whether the test had returned a malaria positive result. Information collected on bed-net availability was limited to LLINs provided by the government to households. The indicators of LLIN availability and use were based on household survey indicators recommended by the Roll Back Malaria Partnership, updated in 2018 [31].

Using the records kept by health workers conducting MSaT, surveyors during the household survey recorded the individual responses to malaria test results and the presence of fever on the day of MSaT.

A team of 30 surveyors was trained to conduct the survey. The data were collected using a handheld device. Sample checks were performed for quality assurance. Informed consent was obtained from all participants. Ethical approval for the study was granted by the Institutional Ethics Committee of the State Health Resource Centre, Chhattisgarh. The dataset was completely anonymised prior to analysis. The minimal dataset is available in S1 File.

### Data analysis

A list of the study variables is provided in S1 File. Cross-tabulation was conducted to provide a descriptive analysis. For key indicators, 95% CIs were reported. To identify the determinants of afebrile malaria, a multivariate analysis was performed using a logistic regression model. A similar model was used to identify the determinants of overall malarial positivity. Logistic regression analysis was used to identify the determinants of LLIN use. Data analysis was performed using STATA 15 software.

## Results

### Socio-demographic profile of the sample

The sociodemographic profiles of individuals surveyed in the six rounds of the survey are given in Table 1. The key characteristics of the samples are similar across all the six rounds. The proportion of scheduled tribes in the total sample is approximately three-fourths in each round. Approximately 60% of the surveyed individuals do not have any formal education. The mean household size is approximately five.

The first round of survey reveals that 90.8% of individuals were tested for malaria during the MSaT drive. The corresponding proportions for the 2nd, 3rd, 4th, 5th, and 6th rounds are 90.4%, 84.1%, 80.3%, 76.6% and 84.1%, respectively. In the first round of MSaT, 72.1% of those tested for malaria report that their test was performed by the Mitanin CHWs. The proportions of Mitanin CHWs in malaria tests conducted in the 2nd, 3rd, 4th, 5th, and 6th rounds are 74.4%, 71.4%, 68.5%, 86.5%, and 83.6%, respectively.

### Prevalence of febrile and afebrile malaria

Of the total individuals covered over the six rounds of the survey, 2870 individuals were febrile at the time of malaria testing. Malaria prevalence or malaria positivity, that is, the proportion of malaria-positive individuals out of those tested, is shown in Table 2. Malaria positivity is

**Table 1. Socio-demographic profile of sample in each round.**

| Characteristics | Sample Profile | | | | | |
|---|---|---|---|---|---|---|
| | Frequency (%) | | | | | |
| | Round-1 | Round-2 | Round-3 | Round-4 | Round-5 | Round-6 |
| N | 16,749 | 18,015 | 17,013 | 17,154 | 17,996 | 15,194 |
| **Sex** | | | | | | |
| Male | 47.9 | 48.6 | 48.2 | 48.4 | 48.3 | 48.6 |
| Female | 52.1 | 51.4 | 51.8 | 51.6 | 51.7 | 51.4 |
| **Age** | | | | | | |
| Below 1 year | 1.4 | 1.2 | 1.2 | 0.9 | 1.5 | 1.6 |
| 1–4 years | 7.8 | 7.6 | 6.3 | 7.4 | 6.9 | 6.4 |
| 5–14 years | 19.2 | 19.9 | 19.7 | 19.4 | 19 | 17.7 |
| 15–48 years | 58.7 | 58.8 | 58.5 | 58.7 | 58.3 | 60.5 |
| 49–59 years | 7.0 | 7.3 | 8.2 | 8.1 | 7.8 | 8.2 |
| 60 years and above | 5.9 | 5.2 | 6.2 | 5.5 | 6.4 | 5.6 |
| **Social Group** | | | | | | |
| Scheduled Tribes | 78.6 | 82.9 | 86.2 | 86.9 | 84.1 | 84.5 |
| Scheduled Castes | 2.0 | 4.7 | 1.6 | 1.0 | 2.4 | 2.8 |
| Other Backward Classes | 18.3 | 10.5 | 11.1 | 9.5 | 12.2 | 11.3 |
| Others | 1.1 | 1.9 | 1.1 | 2.6 | 1.1 | 1.4 |
| **Level of Education** | | | | | | |
| No formal education | 58.2 | 59.9 | 64.9 | 60.2 | 61.0 | 63.0 |
| Primary | 15.3 | 13 | 12.6 | 13.8 | 13.2 | 13.3 |
| Middle school | 10.0 | 9.5 | 8.2 | 9.3 | 9.2 | 8.7 |
| High school | 7.2 | 6.7 | 6.9 | 6.7 | 6.6 | 5.7 |
| Above high school | 9.3 | 10.9 | 7.4 | 10 | 9.9 | 9.2 |
| **Mean Household Size** | 5.7 | 5.6 | 5.4 | 5.4 | 5.3 | 5.1 |

several times higher in the febrile population than in the afebrile population. Malaria positivity rates are similar between men and women. Children aged 1–4 years of age show greater malaria positivity than other age groups. Individuals who belong to scheduled tribes show greater malaria positivity than those from other social groups. Malaria positivity is higher in those without formal education.

The overall malaria positivity in the 1st, 2nd, 3rd, 4th, 5th, and 6th rounds is 5.4% (5.1%–5.9%), 3.0% (2.7%–3.2%), 2.2% (2.0%–2.5%), 2.6% (2.4%–3.0%), 1.7% (1.5%–2.0%) and 1.5% (1.3%–1.7%) respectively. The overall trend shows a decline in malaria positivity, although there is a minor increase in the 4th round (Fig 2).

Fig 2 also shows that the proportion of afebrile individuals among the malaria-positive cases is highest in the 2nd round (47.2%) and lowest in the 4th round (19.6%).

## Determinants of malaria and afebrile malaria

The logistic regression model used to identify the determinants of malaria positivity is shown in Table 3. The odds of malaria are lower among individuals who use LLINs. Individuals who belong to scheduled tribes are more likely to be malaria-positive than members of other social groups. Those without formal education are more likely to have malaria than individuals with middle school or higher education. The odds of having malaria are higher in the 1st round of mass screening than in subsequent rounds. The febrile group has a far greater chance of being malaria-positive than the afebrile group.

**Table 2. Prevalence of malaria by individual characteristics.**

|  | Round-1 (N = 15217) | Round-2 (N = 16291) | Round-3 (N = 14302) | Round-4 (N = 13777) | Round-5 (N = 13787) | Round-6 (N = 12524) |
|---|---|---|---|---|---|---|
| **Fever** | | | | | | |
| Proportion of malaria-positive among febrile individuals | 93.3 | 72 | 42.2 | 33.4 | 65.9 | 43.5 |
| Proportion of malaria-positive cases among afebrile individuals | 1.1 | 1.4 | 0.7 | 0.7 | 0.6 | 0.6 |
| **Sex** | | | | | | |
| Male | 5.4 | 3 | 2 | 2.6 | 1.8 | 1.4 |
| Female | 5.5 | 2.9 | 2.3 | 2.7 | 1.7 | 1.7 |
| **Age** | | | | | | |
| Below 1 year | 6.9 | 1.37 | 0.8 | 1 | 1.8 | 0.9 |
| 1–4 years | 11.9 | 6.1 | 3.03 | 3.4 | 3.2 | 2.1 |
| 5–14 years | 6.7 | 4.1 | 2.5 | 3.3 | 2.5 | 3.2 |
| 15–48 years | 4.3 | 2.4 | 2.1 | 2.4 | 1.5 | 1.2 |
| 49–59 years | 4.5 | 2.1 | 1.8 | 2.3 | 1.1 | 0.6 |
| 60 years and above | 4.8 | 2.7 | 1.2 | 2.5 | 1.1 | 1.2 |
| **Social Group** | | | | | | |
| Scheduled tribes | 6.2 | 3.4 | 2.3 | 3.0 | 2.0 | 1.8 |
| Other social groups | 2.6 | 0.7 | 1.2 | 0.3 | 0.5 | 0.2 |
| **Education** | | | | | | |
| No formal education | 6.6 | 3.4 | 2.4 | 2.9 | 2.0 | 1.8 |
| Primary | 5.5 | 3.4 | 2.7 | 2.7 | 2.2 | 2.4 |
| Middle school | 3.8 | 2.4 | 1.5 | 2.4 | 1.3 | 0.5 |
| High school | 3.6 | 1.3 | 1.2 | 2.0 | 0.9 | 0.7 |
| Above high school | 1.7 | 1.3 | 1.3 | 2.0 | 0.6 | 0.5 |

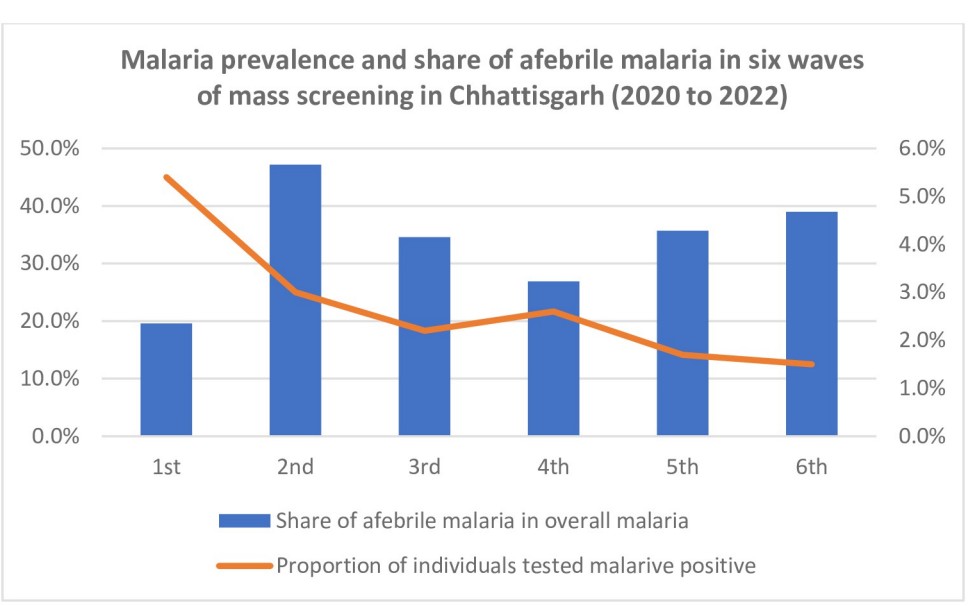

**Fig 2. Malaria prevalence and share of afebrile malaria in six waves of mass screening in Chhattisgarh (2020 to 2022).**

**Table 3. Results of logistic regression models for determinants of malaria and afebrile malaria.**

| Characteristics | All Malaria-positive (N = 84617) | | Afebrile Malaria-positive (N = 81803) | |
| --- | --- | --- | --- | --- |
| | Odds Ratio | p-value | Odds Ratio | p-value |
| **Sex** | | | | |
| Male | 1 | | 1 | |
| Female | 1.07 | 0.20 | 1.07 | 0.34 |
| **Age** | | | | |
| Below 1 year | 1 | | 1 | |
| 1–4 years | 1.58 | 0.15 | 1.90 | 0.13 |
| 5–14 years | 1.50 | 0.20 | 1.69 | 0.21 |
| 15–48 years | 0.97 | 0.93 | 1.21 | 0.64 |
| 49–59 years | 0.70 | 0.28 | 1.00 | 0.99 |
| 60 years and above | 0.75 | 0.40 | 1.33 | 0.52 |
| **Social Group** | | | | |
| Scheduled Tribes | 1 | | 1 | |
| Scheduled Castes | 0.39 | <0.01 | 0.20 | <0.01 |
| Other Backward Classes | 0.47 | <0.01 | 0.34 | <0.01 |
| Others | 0.35 | <0.01 | 0.46 | 0.06 |
| **Education** | | | | |
| No formal education | 1 | | 1 | |
| Primary | 0.88 | 0.13 | 0.90 | 0.36 |
| Middle school | 0.66 | <0.01 | 0.53 | <0.01 |
| High school | 0.68 | <0.01 | 0.58 | 0.01 |
| Above high school | 0.60 | <0.01 | 0.47 | <0.01 |
| **Household Size** | 0.96 | 0.52 | 1.01 | 0.94 |
| **MSaT Round** | | | | |
| Round 1 | 1 | | 1 | |
| Round 2 | 0.76 | <0.01 | 1.32 | 0.01 |
| Round 3 | 0.27 | <0.01 | 0.67 | <0.01 |
| Round 4 | 0.19 | <0.01 | 0.71 | 0.01 |
| Round 5 | 0.38 | <0.01 | 0.55 | <0.01 |
| Round 6 | 0.26 | <0.01 | 0.56 | <0.01 |
| **Used LLIN** | 0.77 | <0.01 | 0.68 | <0.01 |
| **Fever** | 193.27 | <0.01 | - | - |

A similar pattern is found in the association between various factors and afebrile malaria (Table 3). Those from scheduled tribes, without formal education, and not using LLINs are more likely to have afebrile malaria than those from the other corresponding categories of individuals.

## Distribution of LLINs by government and rate of LLIN-use

Overall, 85.8% of households report that they have received at least one LLIN from the government. However, only 37.2% of households have received at least one LLIN for every two individuals. Scheduled tribes, a vulnerable social group have received a similar number of LLINs from the government as rest of the population. Among the households that belong to the scheduled tribes, 87.3% have received at least one LLIN from the government and 37.6% have received at least one LLIN for every two individuals.

The proportion of individuals who report using a LLIN during the previous night is 67.1% (66.4%–67.9%) in the 1st round, 72.7% (72.1%–73.4%) in the 2nd round, 57.7% (57.0%–58.5%) in the 3rd round, 77.0% (76.4%–77.7%) in the 4th round, 55.0% (54.3%–55.8%) in the 5th round and 63.3% (62.61–64.14) in the 6$^{th}$ round.

On average, 9.9% of the surveyed households have not received any LLINs from the government in the three years preceding the survey. Approximately 25.5% of households have received a single LLIN, 45.6% have received two LLINs, and 28.9% of households have received more than two LLINs from the government in the three years preceding the survey.

The results of the logistic regression model used to determine the determinants of LLIN use are presented in Table 4.

Infants were more likely to sleep under LLIN than individuals above five years of age (Table 4). Among the social groups, individuals who belong to scheduled castes are more likely to use LLIN than those in other groups. The use of LLIN increases with educational attainment. LLIN use is higher among individuals who belong to smaller households. Compared to the 1st round, LLIN use is greater in the 2nd and 4th rounds. LLIN use is poorer in winter (3rd

**Table 4. Results of logistic regression for determinants of LLIN-use by individuals (N = 90926).**

| Variable | Odds Ratio | p-value |
|---|---|---|
| **Sex** | | |
| Male | 1 | |
| Female | 1.00 | 0.87 |
| **Age** | | |
| Below 1 year | 1 | |
| 1–4 years | 1.24 | <0.01 |
| 5–14 years | 0.83 | 0.01 |
| 15–48 years | 0.96 | 0.57 |
| 49–59 years | 1.11 | 0.12 |
| 60 years and above | 1.03 | 0.71 |
| **Social Group** | | |
| Scheduled Tribes | 1 | |
| Scheduled Castes | 2.58 | <0.01 |
| Other Backward Classes | 1.67 | <0.01 |
| Others | 0.71 | <0.01 |
| **Education** | | |
| No formal education | 1 | |
| Primary | 1.67 | <0.01 |
| Middle school | 1.66 | <0.01 |
| High school | 1.79 | <0.01 |
| Above high school | 1.93 | <0.01 |
| **Household Size** | 0.58 | <0.01 |
| **MSaT Round** | | |
| Round 1 | 1 | |
| Round 2 | 1.30 | <0.01 |
| Round 3 | 0.63 | <0.01 |
| Round 4 | 1.57 | <0.01 |
| Round 5 | 0.59 | <0.01 |
| Round 6 | 0.86 | <0.01 |
| **No. of LLINs received from the government** | 1.70 | <0.01 |

and 5th rounds) than in the first round. The use of LLIN is higher among individuals whose households receive a greater number of free LLINs from the government.

## Discussion

This study assessed the contribution of afebrile malaria to the overall malaria burden in a region with an extremely high malaria incidence. Although the likelihood of febrile individuals testing positive for malaria was far greater than that of the afebrile population, afebrile malaria still significantly contributed to the overall malaria load. This study found that the share of afebrile malaria in overall malaria cases varied from 19.6% to 47.2% across the six waves of mass screening in 2020, 2021, and 2022. The average proportion of afebrile malaria across the five rounds was approximately 33%. An earlier study based on a smaller sample reported the share of asymptomatic malaria as 57% of the total malaria cases [15]. A study using RDT in a high-burden area of Odisha reported the share of afebrile malaria as 79% [32]. A study on malaria in pregnant women in Chhattisgarh in 2019 reported the share of afebrile malaria as 23.3% [16].

In terms of malaria prevalence in the afebrile population, our study found that malaria positivity varied from 0.6% to 1.4% across the different waves of MSaT, which is similar to a study on pregnant women in Chhattisgarh [16]. However, this is much lower than estimates from studies conducted in various Indian states from 2011 to 2014. These studies reported an afebrile malaria prevalence of 8.4% to 22% [13, 14]. The lower estimate in our study could be due to the declining trend of malaria in Chhattisgarh [33, 34]. One factor that could have helped Chhattisgarh's efforts to prevent and control malaria was the better availability of CHWs in malaria-endemic areas [28]. While other states have one CHW per 1000 population, the density of CHWs in Chhattisgarh is one per 300 people [27, 28]. Mitanin CHWs can maintain intensive contact with the smaller populations they serve. Further, the policies of Chhattisgarh on malaria emphasised training CHWs on malaria and equipping them with RDTs and anti-malarial medicines [28].

The malaria positivity in febrile individuals was above 33% in all six rounds of the study. This indicates that fever continues to be a key symptom of malaria and cannot be neglected when designing strategies and interventions to combat malaria [16].

This study showed that malaria positivity was lower in individuals treated with LLINs. Several studies reported similar global findings [35, 36]. This study showed that LLIN use is associated with a lower prevalence of afebrile malaria. The proportion of individuals using LLINs ranged from 55% to 77% in different rounds. Previous studies in India have reported similar rate of LLIN-use [21, 22, 37]. Our analysis of the determinants of LLIN use showed that individuals who belong to larger households are less likely to use LLIN. Furthermore, our analysis showed that receiving more LLINs from the government increases the likelihood of LLIN use. However, only a quarter of households received more than two LLINs from the government. This is reflected in the low proportion of households (37%) receiving at least one bed net for two members. Therefore, a policy lesson for governments is to ensure better population coverage of the distribution of LLINs in high-burden areas at regular intervals. There are cases of providing more LLINs per household, considering that most households have five to six members. The area covered in this study has a large share of vulnerable communities and poor households, and free distribution by the government can improve equity in accessing LLINs.

Our study showed that scheduled tribes, which are the socioeconomically vulnerable group, were more susceptible to malaria than other social groups. A study conducted in western Kenya also reported that low household income was the most important factor associated with malaria [38]. This highlights the importance of equity in the management of malaria. Malaria

continues to affect the poorest regions and communities worldwide. Governments need to increase efforts targeting vulnerable communities if health equity is to be achieved in the context of malaria.

This study is one of the largest studies on afebrile malaria in terms of the sample size. This was feasible because the MSaT drives were conducted in Chhattisgarh without mass screening and afebrile malaria could not be identified. Chhattisgarh adopted the MSaT approach to combat afebrile malaria. This shows that MSaT can play a key role in improving malaria surveillance, especially for asymptomatic malaria.

Seasonality is an important factor in the context of malaria and multiple rounds of the MSaT allowed us to cover different seasons. Studies on malaria often miss this dimension as many collect data during a single season. Studying LLIN use across different seasons enabled us to offer new insights into this important dimension. LLIN use was the lowest in December, which was the winter season. In India, efforts to promote LLIN use have tended to focus on July, the early monsoon season. These findings highlight the need to promote LLIN use during the winter through behavioural change communication and other strategies.

Our study confirmed the declining trend of malaria infections in Chhattisgarh, but did not attempt to ascertain the impact of the MSaT strategy on reducing malaria prevalence. However, a study in a moderate malaria-burden area in Madhya Pradesh showed the effectiveness of MSaT in reducing the prevalence [39–45]. Further research is recommended to assess the effectiveness of the MSaT strategy in reducing the prevalence of malaria in high-burden areas.

## Limitations

LLIN use was based on self-reporting by respondents, which could carry the risk of overreporting. Information collected on the availability of bed nets in households was limited to LLINs received from the government. Consequently, information on any bed nets that households purchased on their own was excluded, which could result in an underestimation of the availability of bed nets.

The study relied on the recall of household participants, but the responses were recorded using MSaT records. The mean time between an individual being tested in the MSaT and being interviewed was 16 days.

A number of techniques are available for detecting afebrile malaria, including some that are suitable for field conditions [38–45]; however, this study used only one method, RDT.

## Conclusions

This study used the opportunity offered by multiple mass screening drives in Chhattisgarh to examine six repeated cross-sections of household surveys with a large sample. Although fever continues to be a key symptom of malaria, afebrile malaria is a significant contributor in high-burden areas. Global institutions must respond by developing programmes to track and control afebrile malaria. LLINs remain to be a relevant intervention for combating malaria and expanding their free distribution in high-burden populations can aid in attaining the objectives of WHO's HBHI initiative.

## Supporting information

**S1 File. Minimal dataset of the study.**
(DTA)

## Acknowledgments

The authors thank Rohit Yadav for the support in preparing Fig 1.

## Author Contributions

**Conceptualization:** Samir Garg, Ryavanki Sridhar.

**Data curation:** Vishnu Gupta, Mukesh Dewangan.

**Formal analysis:** Samir Garg, Vishnu Gupta, Kavita Patel, Mukesh Dewangan, Prabodh Nanda.

**Funding acquisition:** Samir Garg.

**Investigation:** Vishnu Gupta, Kavita Patel, Mukesh Dewangan.

**Methodology:** Samir Garg, Gajendra Singh.

**Project administration:** Vishnu Gupta.

**Resources:** Kavita Patel, Prabodh Nanda.

**Supervision:** Samir Garg.

**Validation:** Samir Garg.

**Writing – original draft:** Samir Garg, Kavita Patel.

**Writing – review & editing:** Samir Garg, Prabodh Nanda, Ryavanki Sridhar, Gajendra Singh.

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
