## [Decision Letter · Decision Letter 0]

13 Mar 2023

PONE-D-23-02503Assessing afebrile malaria and bed-net use in a high-burden region of India - Findings from multiple rounds of mass screeningPLOS ONE

Dear Dr. Garg,

Thank you for submitting your manuscript to PLoS ONE. After careful consideration, we felt that your manuscript requires substantial revision, following which it can possibly be reconsidered, thus governing the decision of a “major revision”. According to the reviewer # 1 – who has expertise in this field – the authors need to address several concerns, particularly related to the data analysis and methods. Por example, the reviewer complains about a link between results of the test and the symptoms of the participants, especially  if this was collected some weeks after by direct interview with the participants.  

For your guidance, a copy of the reviewers' comments was included below. We therefore invite you to submit a revised version of the manuscript paying close attention to the specific points raised by the reviewer. 

We look forward to receiving your revised manuscript.

Kind regards,

Luzia H Carvalho, Ph.D.

Academic Editor

PLOS ONE

Journal Requirements:

"UNICEF Field Office, Chhattisgarh, India."

We will update your Data Availability statement to reflect the information you provide in your cover letter.\\

5. We note that Figure 1 in your submission contain map image which may be copyrighted. All PLOS content is published under the Creative Commons Attribution License (CC BY 4.0), which means that the manuscript, images, and Supporting Information files will be freely available online, and any third party is permitted to access, download, copy, distribute, and use these materials in any way, even commercially, with proper attribution. For these reasons, we cannot publish previously copyrighted maps or satellite images created using proprietary data, such as Google software (Google Maps, Street View, and Earth). For more information, see our copyright guidelines: http://journals.plos.org/plosone/s/licenses-and-copyright.

Reviewers' comments:

Reviewer's Responses to Questions

**Comments to the Author**

1. Is the manuscript technically sound, and do the data support the conclusions?

Reviewer #1: Partly

2. Has the statistical analysis been performed appropriately and rigorously? 

Reviewer #1: Yes

3. Have the authors made all data underlying the findings in their manuscript fully available?

Reviewer #1: No

4. Is the manuscript presented in an intelligible fashion and written in standard English?

Reviewer #1: Yes

5. Review Comments to the Author

Reviewer #1: The manuscript submitted by Garg et al. describes the results of an observational study on malaria and LLIN use conducted during 6-time points, and it relates these results to the results of six waves of mass screening for malaria parasite carriers in a high-burden region of India. The study is highly powered with more than 15,000 participants in each round. I would like to congratulate the authors for their great effort in this massive study and for wrapping up together an analysis and a manuscript describing these results, which are relevant for understanding malaria epidemiology in India and can be used for malaria control and elimination programmes.

I would like to point out some issues for discussion:

- In my opinion, the main issue of this study is how the information on fever and the results of the tests were collected, like the authors point out in the Limitations section. I find very difficult to link the results of the test with the symptoms of the participants if this was collected some weeks after by direct interview with the participants only. I would like to see some justification in the manuscript to the following questions:

1. Is somehow possible to know the real MSaT results per each of the participants in your survey?

2. Was Body Temperature checked during the MSaT?

3. How was collected the febrile/afebrile information for your survey? Did you ask the participants for fever symptoms or for the Body Temperature register on the test day?

4. What was the mean time between each MSaT round and the household survey?

5. This sentence in the Methods section “The completion of each round of MSaT was followed immediately by a wave of household survey” and the sentence in the Limitations section “the recall period was not more than two months of screening” are somehow vague, I would like to understand this period of time.

- I like the fact that the authors choose to talk about “afebrile malaria”, instead of using “asymptomatic malaria”. As explained in the Introduction section, although not having fever, human population can be clinically affected by this undiagnosed Plasmodium spp parasitaemia such as anaemia, maternal and neonatal mortality, neuronal impairment, or splenomegaly. However, I would like to see the consistent use of this concept all along the manuscript, especially in the Discussion section where the authors switch between the two concepts when describing results of previous studies, and in the Conclusions section.

- In the Introduction section, please define the acronym RDK (Rapid Diagnostic Test Kit).

- In the Introduction section, please define the diagnostic method used in the examples you report on studies checking prevalence. I believe that the study of Ganguly et al. used PCR for diagnostic of parasitaemia while most of the others reported RDT or microscopy results.

- In the Results section, “In the first round of the MSaT, 72.1% of those tested for malaria reported that their test had been performed by the Mitanin CHWs. The share of Mitanin CHWs in the malaria tests conducted in the 2nd, 3rd, 4th, 5th and 6th rounds was….” This reviewer does not understand who did performed the tests that were not done by the Mitanin, if the MSaT was conducted by these CHWs. Does this mean that the test for the rest of participants was conducted by the household survey team or that the test was performed in a health centre as part of a patient visit inside/outside the MSaT strategy? Please, develop this sentence (maybe it can also be done in the Methods section).

- The variable “Number of LLIN received by the Government” gives very few information about LLIN coverage in the households (participants could have bought LLINs aside of the given by Government? Do they conserve the nets at the moment of the survey? are those houses with more inhabitants receiving more nets?). Following the recommendations of the Roll Back Malaria Monitoring and Evaluation group, it would be nice to understand variables of LLIN coverage (1), such as Proportion of households with at least one LLIN, or Proportion of households with at least one LLIN for every two people, or Proportion of population with adequate access (≥ 1 LLIN/2 individuals) to a LLIN in their household.

- Your results report lower parasitaemia prevalence and afebrile malaria than in the nearby Odisha state, being such similar settings. It would be nice to understand the differences in the antimalarial programmes between both states (if any) to understand possible explanations. Also, is any other vector control strategy aside of LLIN distribution (example, IRS) being deployed in the study area?

- In the Discussion section, “Those studies had reported the afebrile malaria prevalence at 4.8% to 22% [13, 14]”. I believe there is a typo error in this sentence, because the reported prevalence of malaria in the Ganguly et al. article is 8.4%, no 4.8%.

- It would be nice to have some discussion on the different techniques to be used for the MSaT strategy or for detecting afebrile population with parasitaemia, effectiveness in detecting submicroscopic parasitaemia or cost-effectiveness of these strategies using molecular tools. If the authors wish, they can find some references to use at the end of this review of studies conducted in India (2, 3) and elsewhere (4, 5, 6, 7, 8).

- In the Discussion section, “The area covered in the current study had a large share of vulnerable communities and poor households and free distribution by government can improve the equity in access to LLINs.” It would be nice to see in the Results if the households with lowest socioeconomical outcome were related with having less LLIN coverage (aside of individual use reported).

- There is missing Journal information in Reference number 12.

- I cannot see in the text where the authors cited references numbered 34-35 and 38-40.

- References for authors’ consideration:

1. Roll Back Malaria Monitoring and Evaluation Reference Group. Household survey indicators for malaria control 2018 [Available from: Household Survey Indicators for Malaria Control_FINAL.pdf (endmalaria.org).

2. Singh A, Rajvanshi H, Singh MP, Bhandari S, Nisar S, Poriya R, et al. Mass screening and treatment (MSaT) for identifying and treating asymptomatic cases of malaria-malaria elimination demonstration project (MEDP), Mandla, Madhya Pradesh. Malar J. 2022;21(1):395.

3. Kaura T, Kaur J, Sharma A, Dhiman A, Pangotra M, Upadhyay AK, et al. Prevalence of submicroscopic malaria in low transmission state of Punjab: A potential threat to malaria elimination. J Vector Borne Dis. 2019;56(1):78-84.

4. Aydin-Schmidt B, Xu W, Gonzalez IJ, Polley SD, Bell D, Shakely D, et al. Loop mediated isothermal amplification (LAMP) accurately detects malaria DNA from filter paper blood samples of low density parasitaemias. PLoS One. 2014;9(8):e103905.

5. Hopkins H, Gonzalez IJ, Polley SD, Angutoko P, Ategeka J, Asiimwe C, et al. Highly sensitive detection of malaria parasitemia in a malaria-endemic setting: performance of a new loop-mediated isothermal amplification kit in a remote clinic in Uganda. The Journal of infectious diseases. 2013;208(4):645-52.

6. Morris U, Khamis M, Aydin-Schmidt B, Abass AK, Msellem MI, Nassor MH, et al. Field deployment of loop-mediated isothermal amplification for centralized mass-screening of asymptomatic malaria in Zanzibar: a pre-elimination setting. Malaria journal. 2015;14:205.

7. Oriero EC, Van Geertruyden JP, Nwakanma DC, D'Alessandro U, Jacobs J. Novel techniques and future directions in molecular diagnosis of malaria in resource-limited settings. Expert Rev Mol Diagn. 2015;15(11):1419-26.

8. Oriero EC, Jacobs J, Van Geertruyden JP, Nwakanma D, D'Alessandro U. Molecular-based isothermal tests for field diagnosis of malaria and their potential contribution to malaria elimination. J Antimicrob Chemother. 2015;70(1):2-13.

6. PLOS authors have the option to publish the peer review history of their article (what does this mean?). If published, this will include your full peer review and any attached files.

Reviewer #1: No

---

## [Author Response · Author response to Decision Letter 0]

25 Apr 2023

Point by point response to reviewer’s comments

Reviewer #1: The manuscript submitted by Garg et al. describes the results of an observational study on malaria and LLIN use conducted during 6-time points, and it relates these results to the results of six waves of mass screening for malaria parasite carriers in a high-burden region of India. The study is highly powered with more than 15,000 participants in each round. I would like to congratulate the authors for their great effort in this massive study and for wrapping up together an analysis and a manuscript describing these results, which are relevant for understanding malaria epidemiology in India and can be used for malaria control and elimination programmes.

Response: We thank the reviewer for the valuable comments and suggestions. We have tried to address all of them. We believe this has helped us in improving the quality of the manuscript. 

I would like to point out some issues for discussion:

- In my opinion, the main issue of this study is how the information on fever and the results of the tests were collected, like the authors point out in the Limitations section. I find very difficult to link the results of the test with the symptoms of the participants if this was collected some weeks after by direct interview with the participants only. I would like to see some justification in the manuscript to the following questions:

1. Is somehow possible to know the real MSaT results per each of the participants in your survey?

Response: The malaria-test results of all individuals were recorded by the health workers conducting MSaT, in the format prescribed by government. During the household survey, surveyors tallied the participants’ responses with the results recorded during MSaT. We have added these details in the Methods section. 

2. Was Body Temperature checked during the MSaT?

Response: At the time of MSaT, body temperature of each individual was checked and fever/no fever was recorded by health workers in the format prescribed by government. During the household survey, our surveyors tallied the participants’ responses with the results recorded during MSaT. We have now mentioned this in the Methods.

3. How was collected the febrile/afebrile information for your survey? Did you ask the participants for fever symptoms or for the Body Temperature register on the test day?

Response: During the household survey, our surveyors tallied the participants’ responses with the results recorded during MSaT.

4. What was the mean time between each MSaT round and the household survey?

Response: We have now calculated the mean difference between the recorded date of testing and the date of interview. The mean time difference was of 16 days. We have mentioned this now in the revised manuscript. 

5. This sentence in the Methods section “The completion of each round of MSaT was followed immediately by a wave of household survey” and the sentence in the Limitations section “the recall period was not more than two months of screening” are somehow vague, I would like to understand this period of time.

Response: We had mentioned the two-month interval as the worst case scenario if the person who got tested at the beginning of MSaT round got interviewed at the end of household survey. We realise it needs to be clarified. 

We have modified the sentence to say that the mean time interval was 16 days. The household survey was completed within a month of each MSaT round. 

- I like the fact that the authors choose to talk about “afebrile malaria”, instead of using “asymptomatic malaria”. As explained in the Introduction section, although not having fever, human population can be clinically affected by this undiagnosed Plasmodium spp parasitaemia such as anaemia, maternal and neonatal mortality, neuronal impairment, or splenomegaly. However, I would like to see the consistent use of this concept all along the manuscript, especially in the Discussion section where the authors switch between the two concepts when describing results of previous studies, and in the Conclusions section.

Response: We have made the suggested changes to use ‘afebrile’ consistently. However, we have retained asymptomatic where the cited study had used the term asymptomatic. 

- In the Introduction section, please define the acronym RDK (Rapid Diagnostic Test Kit).

Response: We have added the definition. 

- In the Introduction section, please define the diagnostic method used in the examples you report on studies checking prevalence. I believe that the study of Ganguly et al. used PCR for diagnostic of parasitaemia while most of the others reported RDT or microscopy results.

Response: We have added the type of test used for each study cited. 

- In the Results section, “In the first round of the MSaT, 72.1% of those tested for malaria reported that their test had been performed by the Mitanin CHWs. The share of Mitanin CHWs in the malaria tests conducted in the 2nd, 3rd, 4th, 5th and 6th rounds was….” This reviewer does not understand who did performed the tests that were not done by the Mitanin, if the MSaT was conducted by these CHWs. Does this mean that the test for the rest of participants was conducted by the household survey team or that the test was performed in a health centre as part of a patient visit inside/outside the MSaT strategy? Please, develop this sentence (maybe it can also be done in the Methods section).

Response: The remaining tests were done by supervisors of CHWS and the government paramedical workers known as Rural Health Organisers (with two-year paramedical diploma). We have added this information in the methods. 

- The variable “Number of LLIN received by the Government” gives very few information about LLIN coverage in the households (participants could have bought LLINs aside of the given by Government? Do they conserve the nets at the moment of the survey? are those houses with more inhabitants receiving more nets?). Following the recommendations of the Roll Back Malaria Monitoring and Evaluation group, it would be nice to understand variables of LLIN coverage (1), such as Proportion of households with at least one LLIN, or Proportion of households with at least one LLIN for every two people, or Proportion of population with adequate access (≥ 1 LLIN/2 individuals) to a LLIN in their household.

Response: The study collected information only on government-given bed-nets i.e. LLINs. We have highlighted this gap in the limitations section now. We have now added the information on the indicators – Households received at least 1 LLIN; and Households received at least 1 LLIN per 2 members. 

- Your results report lower parasitaemia prevalence and afebrile malaria than in the nearby Odisha state, being such similar settings. It would be nice to understand the differences in the antimalarial programmes between both states (if any) to understand possible explanations. Also, is any other vector control strategy aside of LLIN distribution (example, IRS) being deployed in the study area?

Response: IRS is used in both states. The malaria control programmes in the two states may not be very different. One crucial difference is of density of CHWs in malaria endemic areas. In Odisha, the average population per CHW is around 1000 whereas it is one-third of that in Chhattisgarh. This could have helped Chhattisgarh in achieving a better population coverage in malaria prevention and treatment efforts. We have included this dimension in the Discussion now. 

- In the Discussion section, “Those studies had reported the afebrile malaria prevalence at 4.8% to 22% [13, 14]”. I believe there is a typo error in this sentence, because the reported prevalence of malaria in the Ganguly et al. article is 8.4%, no 4.8%.

Response: We have corrected the error. 

- It would be nice to have some discussion on the different techniques to be used for the MSaT strategy or for detecting afebrile population with parasitaemia, effectiveness in detecting submicroscopic parasitaemia or cost-effectiveness of these strategies using molecular tools. If the authors wish, they can find some references to use at the end of this review of studies conducted in India (2, 3) and elsewhere (4, 5, 6, 7, 8).

Response: We have included the suggested reference on household survey indicators and MSaT in Madhya Pradesh. We have included a mention of different available techniques for detecting submicroscopic parasitaemia in the limitations section as this dimension was not covered in the present study.

- In the Discussion section, “The area covered in the current study had a large share of vulnerable communities and poor households and free distribution by government can improve the equity in access to LLINs.” It would be nice to see in the Results if the households with lowest socioeconomical outcome were related with having less LLIN coverage (aside of individual use reported).

Response: The Scheduled tribes were the most disadvantaged social group. They constituted around 85% of the total sample. They received LLINs similar to the overall population. We have added the information.

- There is missing Journal information in Reference number 12.

Response: We have added the information

- I cannot see in the text where the authors cited references numbered 34-35 and 38-40.

Response: We have made the corrections.

- References for authors’ consideration:

1. Roll Back Malaria Monitoring and Evaluation Reference Group. Household survey indicators for malaria control 2018 [Available from: Household Survey Indicators for Malaria Control_FINAL.pdf (endmalaria.org).

2. Singh A, Rajvanshi H, Singh MP, Bhandari S, Nisar S, Poriya R, et al. Mass screening and treatment (MSaT) for identifying and treating asymptomatic cases of malaria-malaria elimination demonstration project (MEDP), Mandla, Madhya Pradesh. Malar J. 2022;21(1):395.

3. Kaura T, Kaur J, Sharma A, Dhiman A, Pangotra M, Upadhyay AK, et al. Prevalence of submicroscopic malaria in low transmission state of Punjab: A potential threat to malaria elimination. J Vector Borne Dis. 2019;56(1):78-84.

4. Aydin-Schmidt B, Xu W, Gonzalez IJ, Polley SD, Bell D, Shakely D, et al. Loop mediated isothermal amplification (LAMP) accurately detects malaria DNA from filter paper blood samples of low density parasitaemias. PLoS One. 2014;9(8):e103905.

5. Hopkins H, Gonzalez IJ, Polley SD, Angutoko P, Ategeka J, Asiimwe C, et al. Highly sensitive detection of malaria parasitemia in a malaria-endemic setting: performance of a new loop-mediated isothermal amplification kit in a remote clinic in Uganda. The Journal of infectious diseases. 2013;208(4):645-52.

6. Morris U, Khamis M, Aydin-Schmidt B, Abass AK, Msellem MI, Nassor MH, et al. Field deployment of loop-mediated isothermal amplification for centralized mass-screening of asymptomatic malaria in Zanzibar: a pre-elimination setting. Malaria journal. 2015;14:205.

7. Oriero EC, Van Geertruyden JP, Nwakanma DC, D'Alessandro U, Jacobs J. Novel techniques and future directions in molecular diagnosis of malaria in resource-limited settings. Expert Rev Mol Diagn. 2015;15(11):1419-26.

8. Oriero EC, Jacobs J, Van Geertruyden JP, Nwakanma D, D'Alessandro U. Molecular-based isothermal tests for field diagnosis of malaria and their potential contribution to malaria elimination. J Antimicrob Chemother. 2015;70(1):2-13.

Editors’s Comments: 

We look forward to receiving your revised manuscript.

Kind regards,

Luzia H Carvalho, Ph.D.

Academic Editor

PLOS ONE

Journal Requirements:

Response: We have modified the file names and style according to PLOS ONE’s guide

Response: We have edited the manuscript for language. 

"UNICEF Field Office, Chhattisgarh, India."

Response: We have added the amended statement and mentioned in our revised cover letter.

We will update your Data Availability statement to reflect the information you provide in your cover letter.\\

Response: We have uploaded the dataset. We have mentioned this in the revised Cover Letter.

5. We note that Figure 1 in your submission contain map image which may be copyrighted. All PLOS content is published under the Creative Commons Attribution License (CC BY 4.0), which means that the manuscript, images, and Supporting Information files will be freely available online, and any third party is permitted to access, download, copy, distribute, and use these materials in any way, even commercially, with proper attribution. For these reasons, we cannot publish previously copyrighted maps or satellite images created using proprietary data, such as Google software (Google Maps, Street View, and Earth). For more information, see our copyright guidelines: http://journals.plos.org/plosone/s/licenses-and-copyright.

Response: We have removed Figure 1.

---

## [Decision Letter · Decision Letter 1]

9 May 2023

PONE-D-23-02503R1Assessing afebrile malaria and bed-net use in a high-burden region of India - Findings from multiple rounds of mass screeningPLOS ONE

Dear Dr. Garg,

After careful consideration, we feel that your manuscript will likely be suitable for publication if the authors revise it to address specific points raised now by the reviewer. According to the reviewers, there are some specific areas where further improvements would be of substantial benefit to the readers.   A major concern is related to English language. At this time, we strongly suggest a professional copy-editing service. For your guidance, a copy of the reviewers' comments was included below. 

We look forward to receiving your revised manuscript.

Kind regards,

Luzia H Carvalho, Ph.D.

Academic Editor

PLOS ONE

Journal Requirements:

Reviewers' comments:

Reviewer's Responses to Questions

**Comments to the Author**

1. If the authors have adequately addressed your comments raised in a previous round of review and you feel that this manuscript is now acceptable for publication, you may indicate that here to bypass the “Comments to the Author” section, enter your conflict of interest statement in the “Confidential to Editor” section, and submit your "Accept" recommendation.

Reviewer #1: All comments have been addressed

2. Is the manuscript technically sound, and do the data support the conclusions?

Reviewer #1: Yes

3. Has the statistical analysis been performed appropriately and rigorously? 

Reviewer #1: Yes

4. Have the authors made all data underlying the findings in their manuscript fully available?

Reviewer #1: Yes

5. Is the manuscript presented in an intelligible fashion and written in standard English?

Reviewer #1: No

6. Review Comments to the Author

Reviewer #1: Dear Authors,

This reviewer is happy with the answers given to the scientific questions raised at the first revision.

However, as suggested by the editorial during the first revision, I think that the manuscript would be benefited of a thorough scientific and English language revision. There are many sentences difficult to read and the text does not flow well at all. It happens with some verb structures and there is an abuse of phrasal verbs. One example is the use of the verb "tally", it is not clear what you mean by this, it has the meaning of "counting" or the meaning of "recording"? Changing some sentences to a more scientific standard language will help your manuscript to flow in a better way.

I suggest to the authors to check similar manuscripts that have been published in this journal:

- https://journals.plos.org/plosone/article?id=10.1371/journal.pone.0232874

- https://journals.plos.org/plosone/article?id=10.1371/journal.pone.0210578

- https://journals.plos.org/plosone/article?id=10.1371/journal.pone.0282209

Other suggestion that you may like to take into account is to add at least one figure to the manuscript, I understand the issue with the copyright of Figure 1. Maybe you would like to built a similar figure by your own so we can understand the geographical setting of the study, or maybe some of the main data can be shown in a barr diagram or something similar.

Thank you.

7. PLOS authors have the option to publish the peer review history of their article (what does this mean?). If published, this will include your full peer review and any attached files.

Reviewer #1: No

---

## [Author Response · Author response to Decision Letter 1]

20 May 2023

Response to reviewer comments:

Reviewer #1: Dear Authors,

This reviewer is happy with the answers given to the scientific questions raised at the first revision.

However, as suggested by the editorial during the first revision, I think that the manuscript would be benefited of a thorough scientific and English language revision. There are many sentences difficult to read and the text does not flow well at all. It happens with some verb structures and there is an abuse of phrasal verbs. One example is the use of the verb "tally", it is not clear what you mean by this, it has the meaning of "counting" or the meaning of "recording"? Changing some sentences to a more scientific standard language will help your manuscript to flow in a better way.

I suggest to the authors to check similar manuscripts that have been published in this journal:

- https://journals.plos.org/plosone/article?id=10.1371/journal.pone.0232874

- https://journals.plos.org/plosone/article?id=10.1371/journal.pone.0210578

- https://journals.plos.org/plosone/article?id=10.1371/journal.pone.0282209

Other suggestion that you may like to take into account is to add at least one figure to the manuscript, I understand the issue with the copyright of Figure 1. Maybe you would like to built a similar figure by your own so we can understand the geographical setting of the study, or maybe some of the main data can be shown in a barr diagram or something similar.

Response: We thank the reviewer for the valuable suggestions. 

We have reviewed and edited the manuscript to improve language and flow. We have got the language reviewed by a senior researcher. 

We have removed the verb “tally” and edited the sentence to improve clarity. 

We have created the map to show the geographical setting of the study using an open-source application (MAPWINDOW) and edited the base maps available from that source. We have added the reference to this software. 

We have included a chart to report the data on malaria positivity and share of afebrile malaria in different rounds of screening.

---

## [Editor Report · Decision Letter 2]

23 May 2023

PONE-D-23-02503R2Assessing afebrile malaria and bed-net use in a high-burden region of India - Findings from multiple rounds of mass screeningPLOS ONE

Dear Dr. Garg,

Thank you for submitting your manuscript for review to PLoS ONE. After careful consideration, we feel that your manuscript will likely be suitable for publication if the authors revise it to address critical points raised previously  by the reviewer.   A major concern is still related to English language. At this time, we strongly suggest a professional copy-editing service.  Finally, it seems to be essential to include figures to represent study settings (there are free software that the authors can use to build their own figure) and also main data. For your guidance, a copy of the reviewers' comments was included below.  

Please submit your revised manuscript by Jul 07 2023 11:59PM. If you will need more time than this to complete your revisions, please reply to this message or contact the journal office at plosone@plos.org. Please include the following items when submitting your revised manuscript:A rebuttal letter that responds to each point raised by the academic editor and reviewer(s). You should upload this letter as a separate file labeled 'Response to Reviewers'.A marked-up copy of your manuscript that highlights changes made to the original version. You should upload this as a separate file labeled 'Revised Manuscript with Track Changes'.An unmarked version of your revised paper without tracked changes. You should upload this as a separate file labeled 'Manuscript'.

We look forward to receiving your revised manuscript.

Kind regards,

Luzia H Carvalho, Ph.D.

Academic Editor

PLOS ONE

---

## [Author Response · Author response to Decision Letter 2]

2 Jun 2023

Response to reviewer comments:

Reviewer Comment: Thank you for submitting your manuscript for review to PLoS ONE. After careful consideration, we feel that your manuscript will likely be suitable for publication if the authors revise it to address critical points raised previously by the reviewer. A major concern is still related to English language. At this time, we strongly suggest a professional copy-editing service. Finally, it seems to be essential to include figures to represent study settings (there are free software that the authors can use to build their own figure) and also main data. For your guidance, a copy of the reviewers' comments was included below. 

Response: 

We have got the manuscript copy edited for language by a professional editing agency (www.editage.com). We have enclosed the editing certificate from the agency. 

We have created the map to show the geographical setting of the study using an open-source application (MAPWINDOW) and edited the base maps available from that source. We have added the reference to this software. 

We have included a chart to report the data on malaria positivity and share of afebrile malaria in different rounds of screening. 

We have used PACE to validate our figures. We have uploaded the valid tif files generated through PACE.

---

## [Decision Letter · Decision Letter 3]

12 Jun 2023

Assessing afebrile malaria and bed-net use in a high-burden region of India - Findings from multiple rounds of mass screening

PONE-D-23-02503R3

Dear Dr. Garg,

We’re pleased to inform you that your manuscript has been judged scientifically suitable for publication and will be formally accepted for publication once it meets all outstanding technical requirements.

Kind regards,

Luzia H Carvalho, Ph.D.

Academic Editor

PLOS ONE

Additional Editor Comments (optional):

Reviewers' comments:

Reviewer's Responses to Questions

**Comments to the Author**

1. If the authors have adequately addressed your comments raised in a previous round of review and you feel that this manuscript is now acceptable for publication, you may indicate that here to bypass the “Comments to the Author” section, enter your conflict of interest statement in the “Confidential to Editor” section, and submit your "Accept" recommendation.

Reviewer #1: All comments have been addressed

2. Is the manuscript technically sound, and do the data support the conclusions?

Reviewer #1: Yes

3. Has the statistical analysis been performed appropriately and rigorously? 

Reviewer #1: Yes

4. Have the authors made all data underlying the findings in their manuscript fully available?

Reviewer #1: Yes

5. Is the manuscript presented in an intelligible fashion and written in standard English?

Reviewer #1: Yes

6. Review Comments to the Author

Reviewer #1: Dear authors, thanks for addressing the previous comments.

In my opinion the manuscript has significantly improved; it now reflects much better the efforts and findings of your research. Congratulations for the work done.

7. PLOS authors have the option to publish the peer review history of their article (what does this mean?). If published, this will include your full peer review and any attached files.

Reviewer #1: No

---

## [Editor Report · Acceptance letter]

10 Jul 2023

PONE-D-23-02503R3 

Assessing afebrile malaria and bed-net use in a high-burden region of India: Findings from multiple rounds of mass screening 

Dear Dr. Garg:

I'm pleased to inform you that your manuscript has been deemed suitable for publication in PLOS ONE. Congratulations! Your manuscript is now with our production department. 

Kind regards, 

on behalf of

Dr. Luzia H Carvalho 

Academic Editor

PLOS ONE